# Incidence, Risk Factors and Prediction of Secondary Hyperparathyroidism in Preterm Neonates under 32 Weeks’ Gestational Age

**DOI:** 10.3390/nu14163397

**Published:** 2022-08-18

**Authors:** Alejandro Avila-Alvarez, Helena Perez Tato, Andrea Sucasas Alonso, Ana Prado Carro, Jesus Fuentes Carballal

**Affiliations:** 1Neonatology Unit, Pediatrics Department, Complexo Hospitalario Universitario de A Coruña, 15006 A Coruña, Spain; 2INIBIC-Health Research Institute of A Coruña, 15006 A Coruña, Spain; 3Pediatric Endocrinology Unit, Pediatrics Department, Complexo Hospitalario Universitario de A Coruña, 15006 A Coruña, Spain

**Keywords:** metabolic bone disease, prematurity, parathyroid hormone, calcium, phosphorus

## Abstract

In preterm newborns, secondary hyperparathyroidism (HPTH) is an underdiagnosed and undertreated entity. Its detection in the context of metabolic bone mineral disease (MBD) screening programs may be important to guide nutritional treatment. We designed a retrospective cohort study to determine the incidence of HPTH in very premature infants. As secondary objectives, we studied the risk factors, morbidities, and biochemical alterations associated with HPTH. A total of 154 preterm newborns ≤32 weeks gestational age (GA) were included. Of these, 40.3% (*n* = 62) presented with HPTH. In the multivariate analysis, independent risk factors for HPTH were cesarean section (OR: 4.00; 95% CI: 1.59–10.06), oxygen during resuscitation (OR: 3.43; 95% CI: 1.09–10.81), invasive mechanical ventilation (OR: 3.56; 95% CI: 1.63–7.77) and anemia requiring transfusion (OR: 2.37; 95% CI: 1.01–5.57). Among the analytical variables, serum calcium (OR: 0.53; 95% CI: 0.29–0.97), serum phosphate (OR: 2.01; 95% CI: 1.39–2.92), vitamin D (OR: 0.96; 95% CI: 0.93–1), and the calcium/creatinine ratio in urine (OR: 0.05; 95% CI: 0.01–0.28) were independently associated with HPTH. The simplified predictive model included GA and calcium/creatinine ratio in urine and demonstrated an AUC of 0.828. We concluded that HPTH is a frequent entity among very premature infants and that further studies are required to determine the role of HPTH in MBD and the clinical applicability of prediction models.

## 1. Introduction

Despite an increase in the survival rates of extremely premature infants, some of the morbidities associated with preterm birth continue to be a relevant problem. Among them, metabolic bone disease (MBD) of prematurity is one of the most prevalent [1,2].

MBD is a bone mineralization defect of multifactorial origin with short- and long-term negative consequences. Since the bone mineralization process takes place fundamentally in the third trimester of pregnancy, prematurity itself is the main risk factor for MBD. Other risk factors include placental conditions that affect nutrient transfer to the fetus, postnatal exposure to drugs such as methylxanthines or steroids, immobilization related to intensive care or mechanical ventilation and prolonged parenteral nutrition [3]. Although well-known, MBD is still a condition difficult to study due to the diversity of criteria for diagnosis and the lack of reliable screening tools [4]. Most neonatal units use a variable combination of biochemical parameters to assess the risk of MBD from the 2nd to the 4th week of life. The combination of an increase in alkaline phosphatase (ALP) >800 IU/L and a decrease in serum phosphorus (P) <5.5 mg/dL has afforded the best diagnostic yield [5,6]. However, even this combination fails to identify the etiopathogenic basis of MBD, which is necessary to guide the most appropriate mineral supplementation in each case [7]. In recent years, other biochemical markers have been studied, such as parathyroid hormone (PTH) and urinary phosphorus and calcium [8,9,10], as well as other imaging tests such as bone ultrasound [11].

PTH is a peptide hormone that is upregulated in response to hypocalcemia. In the preterm newborn, hyperparathyroidism (HPTH) is usually secondary to a low nutritional intake of calcium; therefore, it may be considered a physiological adjustment to normalize extracellular calcium. Although the early detection of HPTH may help clinicians determine its etiology and, therefore, the most appropriate nutritional treatment, this entity is still underdiagnosed, understudied, and undertreated [8]. Moreover, in the calcipenic state, HPTH may further worsen rickets through the promotion of osteolysis, which is clinically relevant.

In this context, we designed a study to determine the incidence of HPTH in very premature infants, defining its perinatal risk factors and its relationship with neonatal evolution, as well as with other serum and urinary biochemical parameters.

## 2. Materials and Methods

An observational retrospective cohort study was conducted in a level III Neonatal Intensive Care Unit (NICU) pertaining to the Spanish national public health system. Preterm newborns of ≤32 weeks’ gestational age (GA) and birth weight ≤1500 g born in the study hospital and admitted to its NICU between 1 January 2015 and 1 January 2021 were included in this study. Those newborns with major malformations or chromosomal abnormalities, as well as those who died before MBD screening, were excluded. The study protocol was approved by the reference Research Ethics Committee (code 2017/360).

Demographic variables, prenatal and obstetric characteristics, data on resuscitation in the delivery room, and main diagnoses/treatments related to prematurity were collected. Biochemical parameters used for MBD screening in the third week of life were registered, including serum values (ALP, P, calcium (Ca), PTH) and urinary values (calciuria, phosphaturia, creatinuria). Concentrations of P, Ca and ALP were determined using standard procedures with the Advia 2400 Analyzer (Siemens Diagnostic Systems, Erlangen, Germany), and 25-OHD and PTH with the Advia Centaur XP Analyzer (Siemens Healthcare Diagnostics, Erlangen, Germany).

HPTH was defined as PTH > 88 pg/mL and severe HPTH as PTH > 180 pg/mL [8]. HPTH patients were compared to non-HPTH patients based on their baseline characteristics, in-hospital evolution and biochemical parameters.

MBD was defined as ALP > 800 IU/L and serum phosphorus <5.5 mg/dL [5]. Bronchopulmonary dysplasia (BPD) was defined as the need for supplemental oxygen therapy for 28 days of life [12]. Only grade ≥2 necrotizing enterocolitis (NEC) according to the Bell classification was considered [13]. Patent ductus arteriosus (PDA) was diagnosed by echocardiography performed according to clinical criteria and treated according to the unit’s protocol. Intraventricular hemorrhage (IVH) was graded according to Volpe [14]. All newborns were screened for retinopathy of prematurity (ROP) and classified according to international guidelines [15]. Intrauterine growth restriction (IUGR) was defined as birth weight < −1.5 z-score, according to Fenton growth charts [16].

A descriptive analysis was performed, expressing quantitative variables as mean ± standard deviation and qualitative variables as absolute value and percentage. The appropriate tests were applied for univariate analysis (Student’s *t*-test or Mann–Whitney U test for quantitative variables and Chi-square or Fisher’s exact test for qualitative variables). *p* values < 0.05 were considered statistically significant. A multivariate analysis was performed using logistic regression to identify independent risk factors for HPTH. Variables prior to HPTH diagnosis that were statistically significant in the univariate analysis were included, as well as those that, although not statistically significant, were considered clinically relevant. Variables with high collinearity were excluded. The area under the ROC curve (AUC) was calculated for the final prediction models. All data were analyzed with the statistical software SPSS version 24 (IBM Corp, Armonk, NY, USA).

## 3. Results

During the study period, 188 preterm infants with a GA < 32 weeks were born at the study center. Thirteen patients were excluded due to major congenital malformations, 11 due to death prior to MBD screening and 10 due to no available PTH levels because of an insufficient blood sample. Finally, a total of 154 patients were included in this study (see flow-chart in Figure 1).

The mean gestational age was 29.7 ± 2.07 weeks and mean birth weight was 1165.62 ± 259.72 g. The mean age at which MBD screening was performed was 15.68 ± 2.10 days. Of the patients studied, 62 (40.3%) presented with HPTH, of which 14 cases (9.1%) were classified as severe HPTH. The main perinatal characteristics and outcomes are described in Table 1. The data corresponding to the biochemical MBD screening values are reflected in Table 2.

Univariate analysis comparison between patients with or without HPTH is shown in Appendix A. In summary, patients with HPTH were younger and apparently sicker, since they required advanced resuscitation and invasive mechanical ventilation (IMV) more frequently. Most of the prematurity-related morbidities were also more frequent in the HPTH group, and the hospital stay was longer.

The multivariate analysis adjusted by GA is shown in Table 3 and Figure 2. Among clinical variables, cesarean section, oxygen during resuscitation, IMV and anemia requiring transfusion remained as factors favoring HPTH. By contrast, the fortification of breastfeeding was protective for HPTH. Based on these results, and together with analytical variables, predictive models for HPTH were developed. Finally, the predictive model with the highest Area Under the ROC curve (AUC) included GA, vitamin D, type of delivery, and urinary Ca/Cr ratio. With this model, an AUC of 0.915 was obtained with a sensitivity of 0.938 and a specificity of 0.872 (Figure 3a). A simplified model considering just GA and urinary Ca/Cr ratio was developed without sacrificing its good performance (AUC 0.828) (Figure 3b).

## 4. Discussion

Currently, HPTH is an underdiagnosed entity, with few studies describing its incidence and relevance among preterm newborns. In this study, we found that more than 40% of infants presented with HPTH at MBD screening in the third week of life, and this is associated with some relevant clinical and analytical variables.

PTH is a peptide hormone produced in the parathyroid gland that plays a fundamental role in the regulation of phosphocalcic metabolism. Its secretion is stimulated by low levels of serum calcium and PTH acts by raising calcemia and reducing phophatemia [17]. Therefore, HPTH does not occur in phosphate-deficient MBD, but in calcium-deficient MBD. In the setting of hypophosphatemia with normal calcemia, which is frequently observed in MBD biochemical screening, PTH may help clinicians elucidate whether the underlying mechanism is a true deficiency of phosphorus (normal PTH) or calcium (elevated PTH) and, therefore, help individualize treatment.

Despite these potential advantages, PTH is frequently missing from MBD screening programs and, therefore, remains underdiagnosed. This was evidenced by the results of a survey of practices across US neonatal units, in which only 1.7% of responders included PTH as a screening test [18].

In our study, we did not find a significant relationship between HPTH and ALP, and, consequently, neither with MBD defined as ALP > 800 IU/L and a decrease in serum phosphorus <5.5 [7]. Although it was physiologically plausible that HPTH was associated with bone turnover and thus elevated ALP, the lack of an association between ALP and PTH had also been observed in previous studies [8]. This suggests that PTH could be a better MBD marker than ALP, especially in the early stages of calcium-deficient MBD, allowing early nutritional correction that would prevent patients reaching advanced stages with bone involvement, in which an elevation of ALP would be found. Moreira et al. previously demonstrated that elevated PTH was a good serological marker for identifying MBD [19]. Moreover, we must not forget that ALP is not bone-specific and can be affected by dysfunctions in other organs. In agreement with our data, Dowa et al. concluded that serum calcium and phosphate were unsuitable for the detection of HPTH and that only high values (>1300 IU/L) of ALP were predictive of HPTH [20].

Chinoy et al., 2019, in an elegant narrative review of MBD, stated that it was surprising that the crucial role of PTH in MBD had received limited attention, and speculated that this fact was in relation to the unavailability of reliable PTH assays decades ago [7]. It is noteworthy that, in a landmark study by Moreira et al., 82 percent of infants with osteopenia upon X-ray were found to have elevated PTH levels [19], highlighting the potential importance of this parameter.

Regarding prenatal risk factors, when correcting for GA, we did not find a statistically significant association with prenatal variables that, a priori, could be theoretically related to elevated PTH (such as IUGR, maternal smoking, or maternal age) [3,17]. Likewise, we did not find a significant association between HPTH and the use of different drugs that had been previously related to MBD (methylxanthines or steroids) [3,6,7]. This suggests that, although related, the pathophysiology of HPTH is different from MBD. However, we did find a statistically significant association, independent of GA, between HPTH and cesarean delivery, requiring oxygen during resuscitation in the delivery room, IMV during admission, and presenting anemia that required transfusion.

Regarding the association between HPTH, birth by cesarean section, and anemia, a possible explanation relating to these factors could be the fact that cesarean-delivered newborns usually have cord clamping earlier than those born by vaginal delivery, which has been associated with lower hemoglobin levels, and, consequently, lower Fe storage [21]. The situation of clinically relevant anemia may favor postnatal growth restriction and thus an increased risk of MBD and/or HPTH [22]. In addition, the situation of iron overload to which a polytransfused newborn is exposed to can have an influence on phosphorus–calcium metabolism, similar to what occurs in cases of hemochromatosis, which was previously described in relation to MBD [6].

Regarding the analytical data, we observed that parameters such as the calcium/creatinine ratio in urine have a strong association independent of GA (OR, adjusted by GA, of 0.052; 95% CI: 0.010–0.275), and, in combination with clinical parameters, allow the prediction of HPTH. Our simplified model considering only GA and the Ca/Cr ratio in urine afforded an adequate correlation with PTH values (AUC of 0.828, sensitivity of 0.082 and specificity of 0.714). This could be useful, not only for the initial diagnosis, but also to monitor the treatment response more closely and less invasively.

There are some limitations in our study, mainly that MBD was not diagnosed by objectively measuring bone mineral content, as it was not clinical practice in our unit. Moreover, in relation to our retrospective design, nutritional mineral intake quantification was not registered and, unfortunately, these data are lacking in our cohort. Strengths include the detailed analytical and clinical data of a homogeneous population.

## 5. Conclusions

In conclusion, HPTH is a clinically relevant entity in prematurity that is often underdiagnosed. The study of different simple clinical and analytical markers might allow clinicians to determine which patients are at higher risk of developing HPTH. Further studies are required to determine its clinical applicability, but, in our opinion, PTH should already be included in MBD screening protocols.

## Figures and Tables

**Figure 1 nutrients-14-03397-f001:**
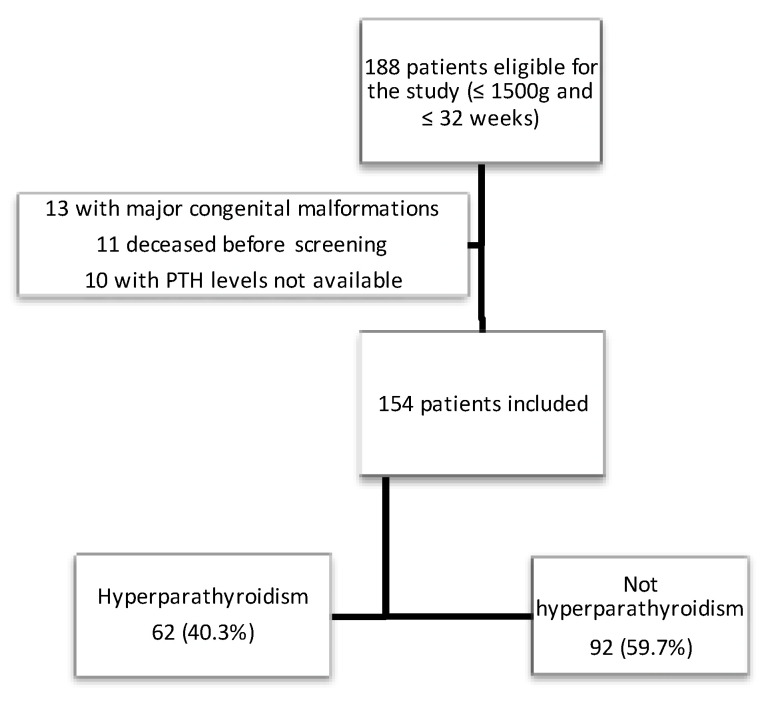
Flow-chart depicting the recruitment of the cohort.

**Figure 2 nutrients-14-03397-f002:**
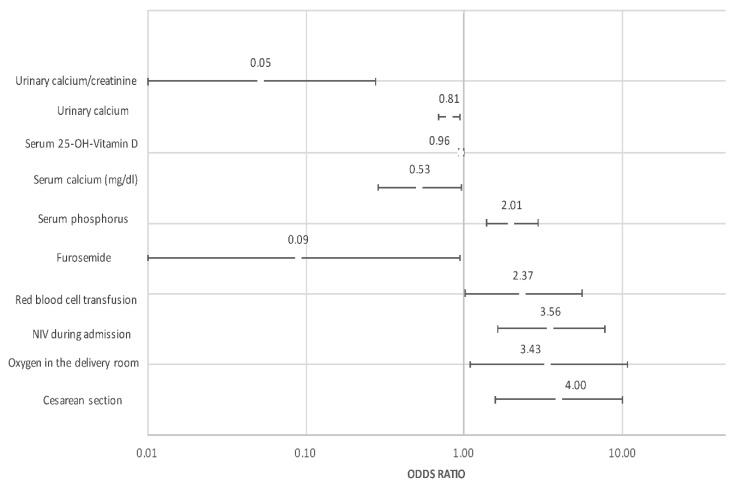
Graph representing odds ratio and 95% confidence interval of factors independently associated with HPTH.

**Figure 3 nutrients-14-03397-f003:**
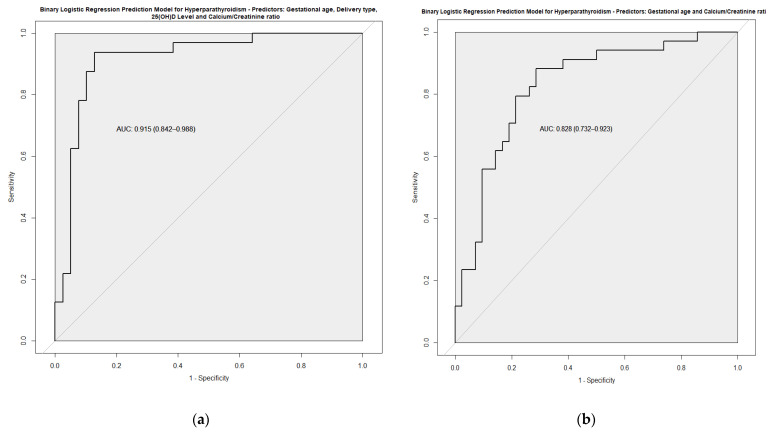
ROC curve of two prediction models developed based on multivariate analysis: (**a**) Model 1; (**b**) Model 2.

**Table 1 nutrients-14-03397-t001:** Perinatal characteristics and main outcomes of the study cohort.

**Perinatal Characteristics**	
$\bar{X}$ ** ± SD**	
Gestational age	29.70 ± 2.07
Maternal age	33.47 ± 6.33
Maternal BMI	25.83 ± 6.14
Apgar minute 1	6.47 ± 1.75
Apgar minute 5	7.89 ± 1.48
Birth weight (g)	1165.62 ± 259.72
Birth weight (z-score)	−0.49 ± 0.76
**No. (%)**	
Maternal diabetes	9 (5.9)
Maternal smoking	18 (11.9)
Maternal arterial hypertension	34 (22.1)
Multiple gestation	58 (37.7)
IVF	37 (24)
Female	81 (52.6)
Antenatal corticosteroids	150 (97.4)
Chorioamnionitis	21 (13.6)
Caesarean section	113 (7.4)
Small for gestational age	29 (18.8)
Ethnicity = caucasian	138 (89.6)
**Neonatal evolution**	
$\bar{X}\;$ **± SD**	
IMV duration (h)	166.80 ± 219.02
Duration of NIMV (h)	198.39 ± 191.61
Duration of oxygen therapy (h)	599.62 ± 655.27
Duration of parenteral nutrition (days)	12.32 ± 7.42
NICU stay (days)	24.71 ± 18.24
Weight at discharge (*n* = 153)	2349.55 ± 379.77
Weight at discharge z-score (*n* = 153)	−1.45 ± 1.17
**No. (%)**	
MBD	22 (14.3)
HPTH	62 (40.3)
Severe hyperparathyroidism	14 (9.1)
Surfactant	70 (45.5)
NIMV during admission	79 (51.3)
IMV during admission	55 (35.7)
Bronchopulmonary dysplasia	40 (26)
Home oxygen	7 (4.5)
Survival without BPD	112 (72.7)
Caffeine	148 (96.1)
Steroids for BPD	18 (11.7)
Furosemide	6 (3.9)
Breastfeeding	134 (87)
Fortification of breastfeeding	124 (80.5)
Death	2 (1.3)
PDA	25 (16.2)
NEC	8 (5.2)
ROP grade ≥ II	7 (5)
Nosocomial sepsis	57 (37)
IVH grade ≥ II	6 (3.9)
PVL	5 (3.2)
Blood transfusion	126 (88.1)

BMI = Body Mass Index; IVF = In vitro fertilization; IUGR = Intrauterine growth restriction; IMV = Invasive mechanical ventilation; NIMV = Non-invasive mechanical ventilation; NICU = Neonatal intensive care unit; MBD = Metabolic bone disease; BPD = Bronchopulmonary dysplasia; PDA = Patent ductus arteriosus; NEC = Necrotizing Enterocolitis; ROP = Retinopathy of Prematurity; IVH = Intraventricular hemorrhage; PVL = Periventricular Leukomalacia; SD = standard deviation.

**Table 2 nutrients-14-03397-t002:** Analytical variables of the study cohort.

$\bar{X}\;$ **± SD**		
Age at screening (days)	15.68 ± 2.10	
Serum creatinine (mg/dL)	0.48 ± 0.20	
Phosphorus in blood (mg/dL)	6.01 ± 1.18	
Calcium in blood (mg/dL)	9.78 ± 0.58	
Blood magnesium (mg/dL) (*n* = 127)	2.05 ± 0.21	
ALP in blood (IU/L)	846.15 ± 351.91	
PTH (pg/mL)	88.29 ± 72.44	
Vitamin D (ng/mL) (*n* = 141)	34.60 ± 11.93	
Phosphorus in urine (mg/dL) *n* = 76	11.86 ± 13.39	
Calcium in urine (mg/dL) *n* = 76	7.46 ± 5.91	
Creatinine in urine (mg/dL) *n* = 77	11.50 ± 9.22	
Ca/Cr ratio in urine *n* = 76	0.92 ± 1.50	
P/Cr ratio in urine *n* = 76	1.26 ± 1.54	
P/Ca ratio in urine (mg/mg) *n* = 75	2.46 ± 4.06	
Tubular phosphate reabsorption (%) *n* = 74	90.77 ± 12.22	

ALP = Alkaline phosphatase; PTH = Parathyroid hormone; Ca = Calcium; P = Phosphorus; Cr = Creatinine.

**Table 3 nutrients-14-03397-t003:** Multivariate analysis to identify independent risk factors for HPTH. Crude Odds Ratios (OR) and OR adjusted by gestational age (GA) are showed.

	OR (95% CI)	OR Adjusted by GA (95%)
**Perinatal variables**		
Maternal body mass index	1.07 (1.01–1.14)	1.05 (0.99–1.12)
Cesarean section	2.24 (1.02–4.91)	4.00 (1.59–10.06)
Oxygen in resuscitation	4.56 (1.49–13.98)	3.43 (1.09–10.81)
Intubation in resuscitation	2.94 (1.20–7.24)	2.18 (0.85–5.59)
**Neonatal Evolution**		
Oxygen	2.47 (1.13–5.39)	1.61 (0.69–3.78)
NIMV	3.12 (1.59–6.14)	2.08 (0.93–4.68)
Invasive mechanical ventilation	4.67 (2.31–9.45)	3.56 (1.63–7.77)
Surfactant	2.97 (1.52–5.79)	2.08 (0.99–4.37)
Patent ductus arteriosus	2.62 (1.09–6.29)	1.45 (0.54–3.92)
Necrotizing enterocolitis	4.82 (0.94–24.72)	3.29 (0.61–17.76)
Anemia requiring transfusion	3.50 (1.65–7.42)	2.37 (1.01–5.57)
Nosocomial sepsis	2.54 (1.29–4.98)	1.91 (0.93–3.90)
ROP Grade > 2	3.97 (1.31–11.98)	2.40 (0.71–8.10)
Bronchopulmonary dysplasia	3.00 (1.43–6.31)	1.66 (0.65–4.29)
Home oxygen	9.75 (1.14–83.12)	5.84 (0.65–52.13)
Breastfeeding Fortification	0.37 (0.16–0.83)	0.30 (0.13–0.70)
Serum phosphorus (mg/dL)	1.59 (1.16–2.17)	2.01 (1.38–2.91)
Serum Calcium (mg/dL)	0.48 (0.26–0.87)	0.52 (0.28–0.96)
Vitamin D (ng/mL)	0.97 (0.94–1.00)	0.96 (0.93–0.99)
Urine phosphorus (mg/dL)	1.03 (0.99–1.07)	1.02 (0.98–1.06)
Urine calcium (mg/dL)	0.87 (0.77–0.98)	0.81 (0.69–0.94)
Calcium/Creatinine Ratio	0.14 (0.03–0.57)	0.05 (0.01–0.27)
Phosphorus/Calcium Ratio	1.22 (0.97–1.53)	1.23 (0.98–1.53)

## Data Availability

All the data are available on reasonable request following publication.

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
