# Peer review of "Incidence, Risk Factors and Prediction of Secondary Hyperparathyroidism in Preterm Neonates under 32 Weeks’ Gestational Age"

_nutrients, 2022, doi:10.3390/nu14163397_

Round 1
Reviewer 1 Report
Reviewer comments
This is an interesting study on Incidence, risk factors and prediction of secondary hyperparathyroidism in preterm neonates under 32 weeks of gestational age with accept for publication.
Comments:-
1- It is well writing study on this topic.
2- The methods were sufficiently documented.
3- The statistical methods are valid and correctly applied.
4- The quality of the figures and tables was satisfactory.
5- Discussion and conclusions were sufficiently concise.
6- The reference list covers the relevant literature adequately.
7- It is a highly interest to a general audience.

Author Response
We would like to thank the external reviewer for their careful and thorough reading of our manuscript and for their thoughtful and positive comments.
Reviewer 2 Report
In this study the incidence of hyperparathyroidism was estimated among very pre-term infants and associated risk factors and comorbidities to HPTH. Overall, the manuscript is of interest and well written. However, some improvements are needed.
Line 40. What the authors mean by “immobilization”?
Overall, the Results section is rather poor. I recommend the authors to add comments on the most relevant findings reported in the Tables, in particular Table 3 and Figure 2.
Table 1. “Survival without BPD” or “survival with BPD”? Also, I think there is a typo in “steroids for BPD” (and not for DBP). As for hyperparathyroidism, the authors should use the already cited acronym “HPTH”.
Line 125. Where may I find out that patients with HPTH are smaller than those without HPTH? Differences in birth weight are not statistically significant.
Lines 158-159. Where was this result reported?
Table 3. The title is incomplete. Multivariate analysis is about what?
Lines 162-163. “This suggests that PTH could be a better MBD marker than ALP”. Why? This point is unclear.
Lines 177-181. Did the authors find a possible explanation for this result?
Author Response
We would like to thank the editorial board and the external reviewer for their careful and thorough reading of our manuscript and for their thoughtful comments and constructive suggestions, which have helped us improve the quality of the manuscript. Please see below our detailed response to each of the reviewer’s comments.
Line 40. What the authors mean by “immobilization”? Thank you for the question. Sometimes, preterm infants on IMV are sedated or even paralyzed, which led to the limitation of spontaneous movements or simply prevent freedom of movement that is necessary to bone turnover and growth. This immobilization related to intensive care has been previously described as a risk factor for MBD. After reviewer´s comment we have specified a bit more the word in the manuscript.
Overall, the Results section is rather poor. I recommend the authors to add comments on the most relevant findings reported in the Tables, in particular Table 3 and Figure 2. Thank you for this suggestion. We tried not to be redundant with tables content, but we agree that we have been too concise. After reviewer´s comment, we have included a brief explanation of the most relevant findings of table 3 and figure 2 and now is as follows:
“The multivariate analysis adjusted by GA is shown in Table 3 and Figure 2. Among clinical variables, cesarean section, oxygen during resuscitation, IMV and anemia requiring transfusion remained as factors favoring HPTH. By contrast, fortification of breastfeeding was protective for HPTH. Based on these results, and together with analytical variables, predictive models for HPTH were developed. Finally, the predictive model with the highest Area Under the ROC curve (AUC) included GA, vitamin D, type of delivery, and urinary Ca/Cr ratio. With this model, an AUC of 0.915 was obtained with a sensitivity of 0.938 and a specificity of 0.872 (figure 3c). A simplified model considering just GA and urinary Ca/Cr ratio was developed without losing its good performance (AUC 0.828) (figure 3b).”
Table 1. “Survival without BPD” or “survival with BPD”? Also, I think there is a typo in “steroids for BPD” (and not for DBP). As for hyperparathyroidism, the authors should use the already cited acronym “HPTH”. Thank you for your comments. The variable is “survival without BPD”. You are right with the typo. We apologize for that, the reason behind the typo is that DBP is the Spanish acronym for BPD. We have modified the table according to your comments.
Line 125. Where may I find out that patients with HPTH are smaller than those without HPTH? Differences in birth weight are not statistically significant. Yes, the reviewer is completely right, there were no statistical differences in birth weight or birth weight z-score. With “smaller” we tried to express differences in GA, but we agree that this is not correct and may be misunderstood. We have changed it for “younger”.
Lines 158-159. Where was this result reported? Thank you for your question. You are right, we have presented the data on the incidence of MBD but not ALP values. Following reviewer´s comment, we have included these data in supplementary table 1. Mean ALP (IU/L) was 815.5±326 in the HPTH group and 890.9 ± 385 in the non-HPTH group (p value = 0.195).
Table 3. The title is incomplete. Multivariate analysis is about what? We agree with the reviewer in this regard. We have modified the title trying to be more descriptive: “Table 3. Multivariate analysis to identify independent risk factors for HPTH. Crude Odds Ratios (OR) and OR adjusted by gestational age (GA) are showed.”
Lines 162-163. “This suggests that PTH could be a better MBD marker than ALP”. Why? This point is unclear. This is a very good point and we have to acknowledge that it was also a matter of debate among authors during the during the writing of the manuscript. Our hypothesis is that ALP is a late biomarker of MBD, as it is only elevated when bone turnover is present. By contrast, PTH may be secreted at earlier stages in response to hypocalcemia. Therefore, at least in calcium-deficient MBD, PTH will be upregulated before ALP.
Following reviewer´s suggestion we have modified our original sentence and now is as follows: “This suggests that PTH could be a better MBD marker than ALP, especially in the early stages of calcium-deficient MBD, allowing early nutritional correction that would prevent reaching advanced stages with bone involvement, in which an elevation of ALP would be found.”
Lines 177-181. Did the authors find a possible explanation for this result? Thank you for your question. We have included a sentence in the manuscript that intended to explain why we did not find an association of HPTH with variables previously reported to be related to MBD: “This suggests that, although related, the pathophysiology of HPTH is different from MBD.”
Probably, the lack of reliable and readily available diagnostic tools for MBD and its complex and heterogeneous pathophysiology prevent us to study risk factors properly in the context of MBD. By contrast, pathophysiology of HPTH is simpler and easier to diagnose and study.